# The Response of Water and Nutrient Dynamics and of Crop Yield to Conservation Agriculture in the Ethiopian Highlands

Sisay A. Belay [1],*, Tewodros T. Assefa [1], P. V. Vara Prasad [2], Petra Schmitter [3], Abeyou W. Worqlul [4], Tammo S. Steenhuis [1,5], Manuel R. Reyes [2] and Seifu A. Tilahun [1]

[1] Faculty of Civil and Water Resources Engineering, Bahir Dar Institute of Technology, Bahir Dar University, Bahir Dar 26, Ethiopia; ttaffese@gmail.com (T.T.A.); tss1@cornell.edu (T.S.S.); satadm86@gmail.com (S.A.T.)

[2] Sustainable Intensification Innovation Lab, Department of Agronomy, Kansas State University, Manhattan, Kansas, KS 66506, USA; vara@ksu.edu (P.V.V.P.); manneyreyes@ksu.edu (M.R.R.)

[3] International Water Management Institute, Yangon 11081, Myanmar; p.schmitter@cgiar.org

[4] Blackland Research Center, Texas A&M AgriLife Research, Temple, TX 76502, USA; aworqlul@brc.tammus.edu

[5] Department of Biological and Environmental Engineering, Cornell University, Ithaca, NY 14850, USA

* Correspondence: sisayasress@gmail.com

**Abstract:** Smallholder agriculture constitutes the main source of livelihood for the Ethiopian rural community. However, soil degradation and uneven distribution of rainfall have threatened agriculture at present. This study is aimed at investigating the impacts of conservation agriculture on irrigation water use, nutrient availability in the root zone, and crop yield under supplementary irrigation. In this study, conservation agriculture (CA), which includes minimum soil disturbance, grass mulch cover, and crop rotation, was practiced and compared with conventional tillage (CT). We used two years' (2018 and 2019) experimental data under paired-t design in the production of a local variety green pepper (*Capsicum annuum* L.). The results showed that CA practices significantly ($\alpha = 0.05$) reduced irrigation water use (13% to 29%) and runoff (29% to 51%) while it increased percolated water in the root zone (27% to 50%) when compared with CT practices under the supplementary irrigation phase. In addition, CA significantly decreased $NO_3$-N in the leachate (14% to 44%) and in the runoff (about 100%), while $PO_4$-P significantly decreased in the leachate (33% to 50%) and in the runoff (16%) when compared with CT. Similarly, CA decreased the $NO_3$-N load in the leachate and in the runoff, while the $PO_4$-P load increased in the leachate but decreased in the runoff. The yield return that was achieved under CA treatment was 30% higher in 2018 and 10% higher in 2019 when compared with the CT. This research improves our understanding of water and nutrient dynamics in green pepper grown under CA and CT. Use of CA provides opportunities to optimize water use by decreasing irrigation water requirements and optimize nutrient use by decreasing nutrient losses through the runoff and leaching.

**Keywords:** conservation agriculture; leachate; conventional tillage; nutrient dynamics; supplementary irrigation

## 1. Introduction

Water and soil nutrients remain the most limiting resources for agriculture. However, rainfed and irrigated agriculture is often hindered by the depletion of soil nutrients through surface runoff and leaching during rainy phases [1,2], as well as by the scarcity of water during dry phases [2,3]. Surface runoff adversely affects the availability of water [3–5], soil nutrients [5–7], and soil organic

matter for plant growth and development [8,9]. Percolated water (leaching) can also affect water and nutrient availability to some extent [5]. Agricultural activities exacerbate the removal of nutrients by either of the processes [10]. Nutrients that are removed by surface runoff are permanently lost before reaching the root zone of the plant, while nutrients that are leached below the root zone are at least temporarily lost from the root system. Concurrently, the nutrient and water components of surface runoff and percolation may also deteriorate the water quality of wells, reservoirs, and lakes [11]. Thus, the nutrients often removed by water movement and dynamics contribute not only to water quality deterioration, but this also imply an economic loss of soil fertility to the farmer. The current approach of agricultural systems, which promotes the use of more chemical fertilizers [12], particularly for vegetable production, shows a wider expansion of the above risks and is becoming a serious threat to our environment [12–14].

Hence, there is a need for a paradigm shift to improve smallholder agriculture systems that promote sustainable intensification, which encourages an increase in crop productivity with minimum inputs and protects the environment at the same time [12]. Smallholder vegetable production at home gardens is one approach of a localized strategy to improve the livelihood and nutrition of farmers in many developing countries [15,16], including Ethiopia. Smallholder vegetable production may also be optimized by applying conservation agriculture (CA) practices that would improve productivity with minimum inorganic inputs and minimize adverse effects on the environment. The CA system (minimum soil disturbance, complete soil cover, and proper crop rotation) has been used to improve irrigation water use efficiency and crop productivity while controlling soil nutrient losses caused by various factors [16,17]. No-tillage, despite the challenges of implementation and adoption [18], reduces runoff [4,19,20], increases percolation, and enhances water holding capacity of soils [7,21–23], when combined with grass mulch cover and proper crop rotation. The biological decomposition of grass mulch has improved the soil quality (adding soil nutrients) and soil structure, while a no-till practice combined with a complete soil cover reduces the soil compaction in the long term [24], particularly in drier regions or dry phases [7,23]. The yield of cereal crops has been increased in CA systems (due to higher water infiltration) under rainfed phases of production [7,19,22,25]. Water use efficiency and the yield of vegetables significantly increase under CA in dry irrigation phases of production compared with the conventional practices [16,26,27].

Most of the previous studies evaluated the impacts of CA practices mainly on cereal crops and on either rainfed or irrigated systems. There are few studies on the impacts of CA on water saving and yield measurement in grain crops in the sub-humid Ethiopian highlands [3,6,14,18,27]. However, experimental field measurements on commercial home vegetable production systems with measurements on water and nutrient dynamics in addition to yield have been limited. Such targeted experiments are required for a comprehensive impact analysis of CA under irrigated-rainfed vegetable production systems. Application of CA in vegetable production systems with local varieties based on market demand has greater opportunities for adoption. The demand for green pepper (*Capsicum annum L.*) is becoming greater, and farmers in this region are choosing to grow in their home gardens. Moreover, there is limited research under the supplementary phase (irrigated and then rainfed) on vegetables and particularly in peppers in Ethiopia. There were some limited to only dry phase production under irrigation; studies on vegetable fields were not continuous and insufficient, and none focused on green peppers, which require well-drained soils or climate conditions to escape from disease. Thus, the objective of this study is to investigate the impacts of CA on water dynamics (runoff, percolation, irrigation) and nutrient (nitrate and phosphorous removals) dynamics, and its contribution to improve crop (local green pepper, *Capsicum annuum* L.) yields under supplemental irrigation and rainfed systems. The results from this study contribute to the comprehensive evaluation and understanding of the CA system for improving productivity, water quality, ecosystem services, and the livelihood of people.

## 2. Materials and Methods

### 2.1. Description of the Study Area

The study area is located in the Dengeshita experimental site in the headwaters of the Blue Nile in the Northern Ethiopian highlands (11.32° N and 36.85° E at an altitude of 2042 m), 80 km south of Bahir Dar, the capital of the Amhara region (Figure 1). Dengeshita has a gentle slope (2% to 5%) with a mean annual rainfall of 1400 mm and a temperature of 18 °C based on data from the Ethiopian National Meteorological Agency, Bahir Dar District. The top 40 cm soil has a loam soil texture, and the interplot variation of soil texture was insignificant based on initial soil analysis results. The soil is slightly acidic with a pH level of 6. Field capacity, permanent wilting point, available soil water, bulk density, total nitrogen, available phosphorus, and available potassium in the top 40 cm soil layer were 0.31 cm$^3$ cm$^{-3}$, 0.21 cm$^3$ cm$^{-3}$, 0.10 cm$^3$ cm$^{-3}$, 1.32 g cm$^{-3}$, 0.93 g kg$^{-1}$, 9.57 mg kg$^{-1}$, and 191 mg kg$^{-1}$, respectively. Detail soil characteristics of the study site can be found in Belay et al. [27].

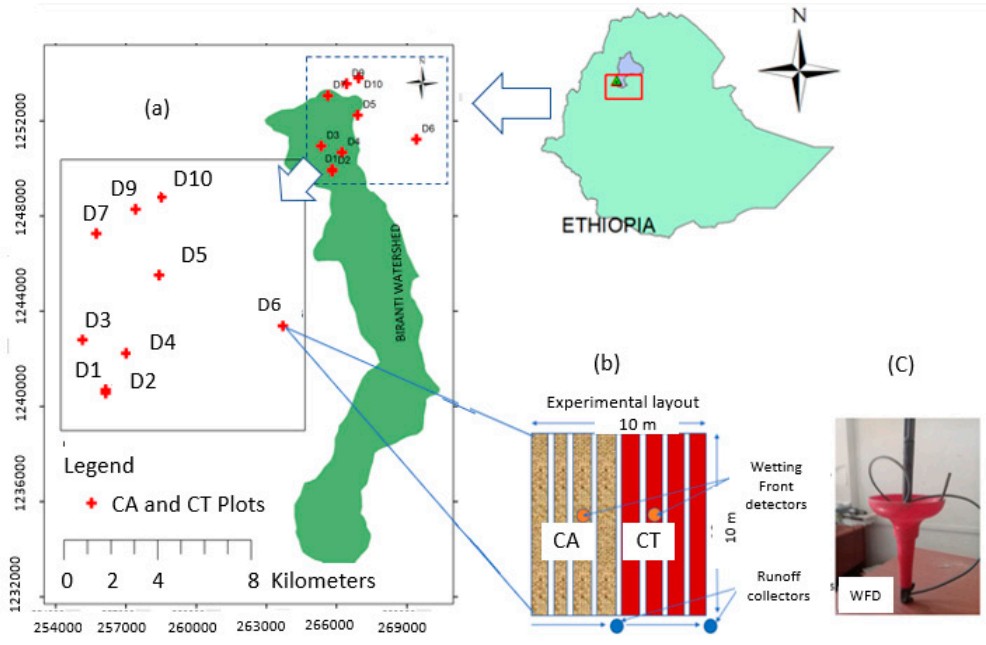

**Figure 1.** Location map of the experimental site (**a**) and layout of conservation agriculture (CA) and conventional tillage (CT) treatments (**a,b**) and wetting front detector (WFD) (**c**). The bottom arrows (**b**) indicate the direction of runoff flows relative to the runoff collector.

### 2.2. Experimental Design and Layout

A total of 10 experimental plots were established on 100 m$^2$ in size, where 50 m$^2$ was randomly assigned for conservation agriculture (CA) and another 50 m$^2$ for conventional tillage (CT) practice under a supplementary irrigated phase (Figure 1a,b). The experimental plots were initially selected based on the availability of productive shallow groundwater wells adjacent to irrigable farms and farmers' willingness to participate in the experiment. CA consists of no-tillage and the application of grass mulch at the rate of 2 t ha$^{-1}$, while CT is the current farmers' practice of 4–6 tillage frequencies (tillage depth 15–25 cm using animal drawn plough) and without mulch cover. A paired-t design was used to examine the impacts of CA on water use, runoff, leachate, nutrient use, and crop yield as compared to CT treatment. Irrigation water was managed by the estimated reference evapotranspiration based on the methods explained by Allen et al. [28]. The crop rotation (onion–pepper–garlic–pepper–onion–pepper) was the same for both CA and CT agricultural practices; however, only the pepper production period was used for this research paper. Drip irrigation was used for both

2018 and 2019 experimental years (from March to mid of June). Each treatment subplot was subjected to an equal amount of irrigation water for a week to ensure uniform recovery of transplanted seedlings.

### 2.3. Crop Management Practices

Local variety pepper (*Capsicum annuum* L.) was transplanted on 13 March in 2018 and 19 March in 2019 (Table 1). The spacing between rows and plants during transplanting was 40 cm. After transplanting, the initial stage lasted for 20 days; the vegetative stage lasted for 30 days; the mid-season growth stage (flowering and fruiting) lasted for 50 days; and the late-season stage lasted for 60 days. Inorganic fertilizer (diammonium phosphate (DAP) and urea) was not applied in the pepper growing period (the 2nd irrigation season) based on the local practice. However, urea fertilizer (46-0-0: N-P-K) was applied to the plots at the rate of 200 kg ha$^{-1}$ using split application method (twice) during the 1st irrigation season from October to March. Sufficient phosphorus fertilizer was available in the soil based on a soil laboratory investigation [27]. The nutrient content of local grass and cow dung was analyzed at the soil laboratory. Local grass (85% organic matter; 0.18% total nitrogen, and 17 ppm available phosphorus,) was applied at the rate of 2 t ha$^{-1}$ as mulch cover for only CA treatment twice per irrigation period. Compost of cow dung or manure (42% organic matter; 2.1% total nitrogen, and 82 ppm available phosphorus) was applied at the rate of 5 t ha$^{-1}$ equally for both treatments at the end of the 1st season harvest. The harvesting period was from June to August for both 2018 and 2019. Farmers used 4 to 6 harvests in each season, and a fresh pepper yield was then weighed from each subplot during every harvest and converted to t ha$^{-1}$.

**Table 1.** Experimental activities of 2018 and 2019 green pepper cropping seasons.

| Year | Crop | Management Activities | Date | Methods and Tools |
|---|---|---|---|---|
| 2018 | Local variety pepper (*Capsicum annuum* L.) | Seedling | 20 January 2018 | Watering-can |
| | | Cow dung application | 5 February 2018 | Manual |
| | | Tillage | 10–20 February 2018 | Draught animal |
| | | Planting | 13 March 2018 | Manual |
| | | Mulch application | 12 March 2018 | Manual |
| | | Irrigation | 13 March 2018–12 May 2018 | Drip irrigation |
| | | Weeding/hoeing | 20 April 2018, 5 May 2018, 10 July 2018 | Handpick |
| | | Harvesting | 1 June 2018–25 August 2018 | Handpick |
| 2019 | Local variety pepper (*Capsicum annuum* L.) | Seedling | 9 January 2018 | Watering-can |
| | | Cow dung application | 5 February 2018 | Manual |
| | | Tillage | 15–25 February 2019 | Draught animal |
| | | Planting | 19 March 2019 | Manual |
| | | Mulch application | 12 March 2019 | Manual |
| | | Irrigation | 19 March 2019–18 May 2019 | Drip irrigation |
| | | Weeding/hoeing | 25 April 2018, 15 May 2018, 15 July 2019 | Handpick |
| | | Harvesting | 10 June 2019–29 August 2019 | Handpick |

### 2.4. Data Collection

Climate data used for calculating the reference evapotranspiration (ET$_o$) with the FAO Penman–Monteith equation [28] were collected from Dangila weather station (15 km from the site) for the period of 1995–2016. We excluded the years 1998–2000 from the period because of the large amount of missing data. We used the average of these processed climate data, which include temperature (maximum and minimum), relative humidity, actual sunshine hours, and wind speed. Crop water use (ET$_c$) was determined by multiplying ET$_o$ by the crop coefficient [28] for the initial, development, mid-season, and end stages. The same crop coefficient was used for the growth stages of pepper crop for the experimental years (i.e., 0.7 for initial, 0.95 for development, 1.05 for mid, and 0.7 for the late season). Irrigation water to be applied to the pepper was determined at an allowable constant soil moisture depletion fraction ($p = 0.4$) of the total available soil water (TAW), where TAW was determined from the permanent wilting point, field capacity, root depth, and bulk density variables. The depth of water applied during each irrigation event was the net irrigation requirement estimated by the Penman–Monteith method (using the long-term data from 1995 to 2016), and that which was needed

for inefficiencies in the irrigation system. Considering conveyance and other losses for a drip system, an irrigation efficiency of 90% was assumed. The same amount of water (5 mm per irrigation on average) was applied in the 2018 and 2019 years with different irrigation scheduling. Irrigation was ceased immediately after the onset of rainfall in mid of May.

Runoff was measured using runoff collectors of a geomembrane sealed trench of size 0.5 m by 0.4 m by 1 m (200 L in capacity), installed at the end of each treatment bed (Figure 1b). One trench was used for a treatment where the runoff drains from 4 beds into the trench. It was recorded during every storm during daytime, and runoff collected during the nighttime was recorded in the morning. Each time after measurement, the trench was cleaned from incoming sediments. The amount of leachate was monitored every 10 days using a wetting front detector (WFD) installed 40 cm below the soil surface (Figure 1c), and evaporation loss for irrigated fields (wet) at this depth was neglected. Capillary rise of water from a 6–10 m water table through a sand filter (always wet for irrigated fields) was also assumed unrealistic. Water passing the fine sand filter was collected at the bottom of the WFD where a small hose was attached to it for draining out the leachate every 10 day using a syringe. The amount of leachate (mL) obtained in the area of the WFD was converted to millimeters of leachate by dividing the cross-section area (20 cm diameter) of the WFD. A water sample of 50 mL (20 mL for $NO_3$-N, 10 mL for $PO_4$-P) was collected from the runoff, as well as the leachate for determining the concentration of nutrients (i.e., $NO_3$-N and $PO_4$-P). Available phosphorus and $NO_3$-N concentrations were determined using the Palintest photometer 7500 tests. The nitrate-nitrogen and available phosphorus loads were calculated by multiplying drainage volumes for each period with the corresponding measured $NO_3$-N and $PO_4$-P concentrations.

Total water used by the crop plus evaporation was calculated using the Penman–Monteith method as stated by Belay et al. [27]. Actual crop water used by the pepper for the growing season was computed using the soil water balance equation by Kresović et al. [29], as shown below:

$$Ta = I + Rf + Cr - R_o - P_{40} \pm \Delta S$$

where $ET_a$ is evapotranspiration (mm) during the growing season; I is the amount of irrigation water applied (mm); $Rf$ is actual rainfall recorded at site (mm); $Cr$ is the capillary rise (mm), considered to be zero because the groundwater table was >4 m below the surface in the growing months; $P_{40}$ is percolation (mm), considered to be at a 40 cm soil depth because the soil water content below 40 cm reached field capacity during rainy season months on the sampling dates; $R_o$ is runoff (mm), measured using runoff collectors because the field was saturated in rainy months (June to August), and $\Delta S$ is the change in soil moisture content (mm) measured using the gravimetric method at the time of transplanting and after harvest.

### 2.5. Data Analysis

All data are presented with arithmetic means and was statistically analyzed using a paired-t analysis for means after checking the normality using the Jarque–Berra method [30]. Phosphorus concentration data were transformed to natural logarithm to observe the normality. All the results shown in the tables and figures are means of treatment plots or replicates. Mean values were compared for any significant differences using the least significant difference (LSD $_{\alpha = 0.05}$) method.

## 3. Results

### 3.1. Irrigation and Rainfall Contributions in the Pepper Growing Period

The amount of rainfall during the growing period of pepper was 594 mm in 2018 and 618 mm in 2019. The contribution of irrigation in 2018 was 46% in the CA (370 mm) and 56% in the CT (476 mm); while in 2019 its contribution was 37% in the CA (255 mm) and 42% in the CT (288 mm) (Table 2). The pepper growth was supported by rain in the wet period (Figure 2). Irrigation water was significantly reduced ($p < 0.05$) under CA compared to CT management. The grass mulch and

no-tillage practices under CA treatment decreased the interval of irrigation application and hence reduced irrigation water. The cumulative depth of irrigation application throughout the dry periods of the pepper growing period is plotted in Figure 2.

**Table 2.** Mean (±standard deviation) for applied irrigation, runoff, percolation, crop water use ($ET_c$), and fresh yield under conservation agriculture (CA) and conventional tillage (CT) treatment in the 2018 and 2019 supplementary irrigated pepper production period.

| Variables | 2018 | | 2019 | |
|---|---|---|---|---|
| | **CA** | **CT** | **CA** | **CT** |
| Applied irrigation (mm) | 367.3 ± 55.4 [b*] | 475.4 ± 68 [a] | 254.8 ± 42.9 [b] | 287.8 ± 57.7 [a] |
| Rainfall (mm) ** | 594.0 | 594.0 | 618.0 | 618.0 |
| Runoff (mm) | 53.2 ± 8.0 [b] | 80.1 ± 22.3 [a] | 95.5 ± 18.4 [b] | 123.6 ± 19.2 [a] |
| Percolation / leaching (mm) | 7.5 ± 2.4 [a] | 5.9 ± 1.9 [b] | 6.9 ± 1.6 [a] | 4.6 ± 2.4 [b] |
| Crop water used ($ET_c$) (mm) | 796.7 ± 65.3 [b] | 855.6 ± 83.1 [a] | 678.7 ± 55.0 [a] | 686.6 ± 69.1 [a] |
| Fresh yield (t ha$^{-1}$) | 11.7 ± 5.9 | 9.1 ± 4.3 | 6.2 ± 1.7 | 5.9 ± 2.3 |
| Contribution of irrigation | 46% | 56% | 37% | 42% |

* Numbers followed by the same letters under the same row heads in the same year are statistically nonsignificant at the $\alpha = 0.05$ significant level. ** Rainfall is assumed the same for the village.

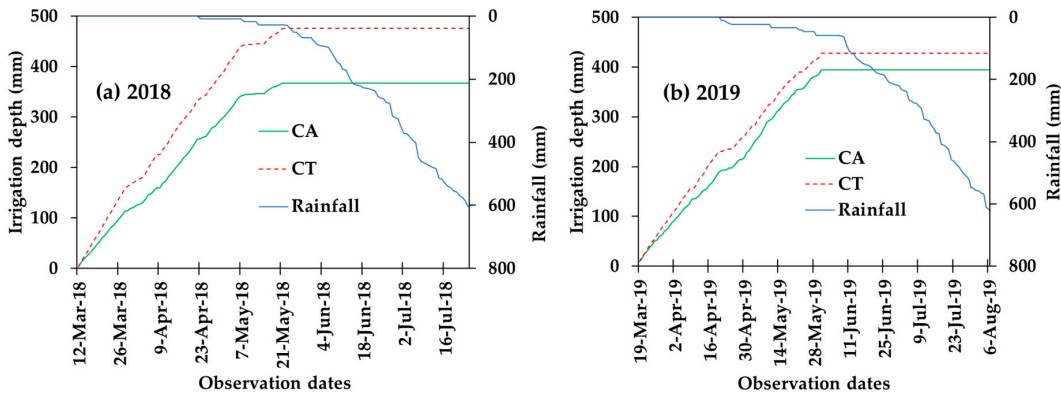

**Figure 2.** Cumulative rainfall versus irrigation water applied for conservation agriculture (CA) and conventional tillage (CT) for the experimental years of 2018 (**a**) and 2019 (**b**).

Year to year difference in the contribution of irrigation application was due to the difference in the time of transplanting pepper. To avoid drainage problems in the rainy period, farmers have practiced the transplanting of pepper at the beginning of March where the initial and development stages occurred in the drier months and the fruit stage occurred in the wet months (Table 1).

### 3.2. Effect of Conservation Agriculture on Water Dynamics (Runoff and Percolation/Leaching)

The average runoff depth for the 2018 and 2019 years were, respectively, 53 and 96 mm under CA and 80 and 124 mm under CT (Table 2). In 2018, a significant ($p < 0.05$) decrease in the average runoff depth (51%) was observed in CA (53 mm) as compared to the CT (80 mm) (Table 2). Similarly, in 2019, a significant ($p < 0.05$) decrease in runoff depth (29%) was observed in CA (96 mm) as compared to the CT (124 mm) (Table 2). The commutative of daily runoff and the rainfall (mm) records are plotted for both years in Figure 3. It shows that the runoff under CT is significantly greater compared to that under CA.

Overall, runoff in year 2019 was significantly greater than that of 2018 (Table 2) because of the difference in the planting date (Table 1) and the delays in the onset of the wet period, which can be related to a lower contribution of irrigation in 2019.

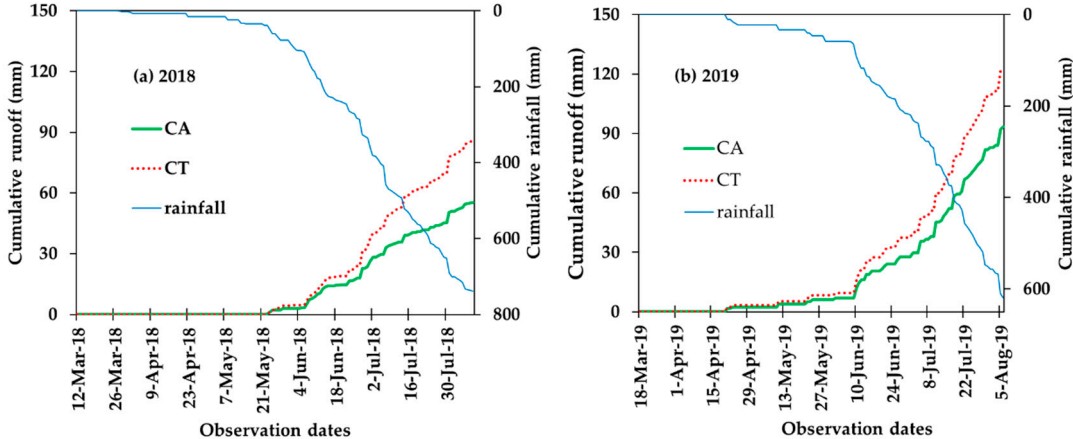

**Figure 3.** Comparison of cumulative daily runoff depth between CA and CT treatments for 2018 (**a**) and 2019 (**b**) during the growth period of pepper.

On the other hand, the quantity of leachate was significantly increased in the CA (8 mm) when compared to the CT (6 mm) for the 2018 experimental season (Table 2; Figure 4). Similarly, in 2019, a significant ($p < 0.05$) increase in leachate was observed in CA (7 mm) as compared to the CT (5 mm) (Table 2; Figure 4). The amount of leachate increased slowly during the dry season while it increased rapidly after the onset of the rainfall around the beginning of May (Figure 4). This supports the nature of complementary processes of runoff and percolation, where a decrease in the former corresponds to an increase in the later. The temporal variation shows that the maximum leachate depth occurred after 20 July over the growing periods and then started decreasing onwards (Figure 4).

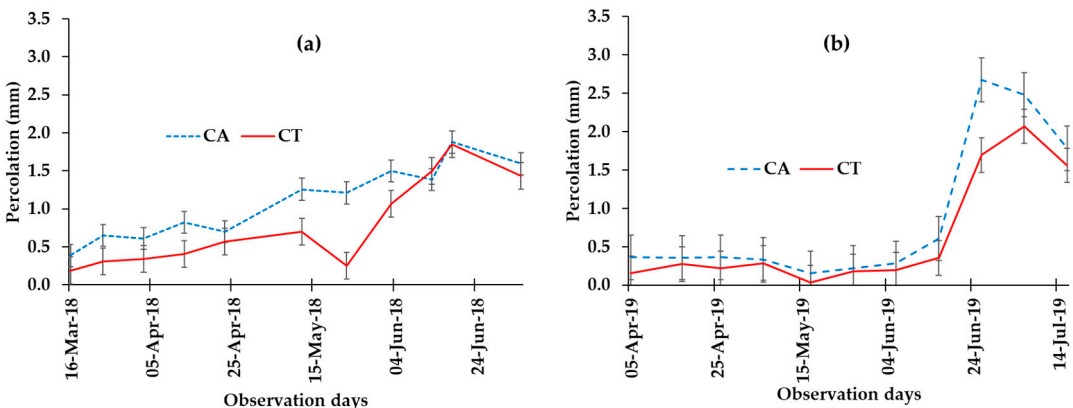

**Figure 4.** Percolated water (leachates) measured every 10 days of the experimental periods of 2018 (**a**) and 2019 (**b**) for CA and CT management.

### 3.3. Consumptive Water Use of Pepper (ET$_c$) under Supplementary Irrigation

The water used by pepper from irrigation and rainfall, i.e., the actual evapotranspiration (ET$_a$), was in the range of 750 to 950 mm and 600 to 850 mm, respectively in the years 2018 and 2019 (Figure 5). The maximum ET$_a$ in the CA management was 770 and 700 mm while it was 798 and 715 mm in CT, respectively, for the years 2018 and 2019. However, the difference between treatments in ET$_a$ was significant ($p < 0.05$) only for 2018 (Table 2). The yield of pepper was significantly greater in CA compared with CT management (Table 2). The average yield of pepper under CA was 11.7 t ha$^{-1}$ in 2018 and 6.2 t ha$^{-1}$ in 2019 while the yield was 9.1 t ha$^{-1}$ in 2018 and 5.9 t ha$^{-1}$ in 2019 under CT management. This shows that the yield achieved in 2018 under CA treatment was 30% higher compared to that of CT. In 2019, the yield of pepper was only 10% higher under CA compared to the CT. The yield difference was statistically different ($p < 0.05$) only for the 2018. The peak pepper

yield in CA occurred ahead of CT management in response to lower optimum water use for the site conditions. However, the yield under CT management has continued even after the end of the last harvest of pepper (Figure 5). The yield results in Figure 4 and the runoff results in Table 2 agree in that runoff in 2018 was less than 2019; however, the yield in 2018 was higher when compared with the yield in 2019. This means that plots in 2019 were subjected to water logging problems. Based on farmers' intrinsic knowledge, which is in line with [31], well-drained soil is suitable to pepper production.

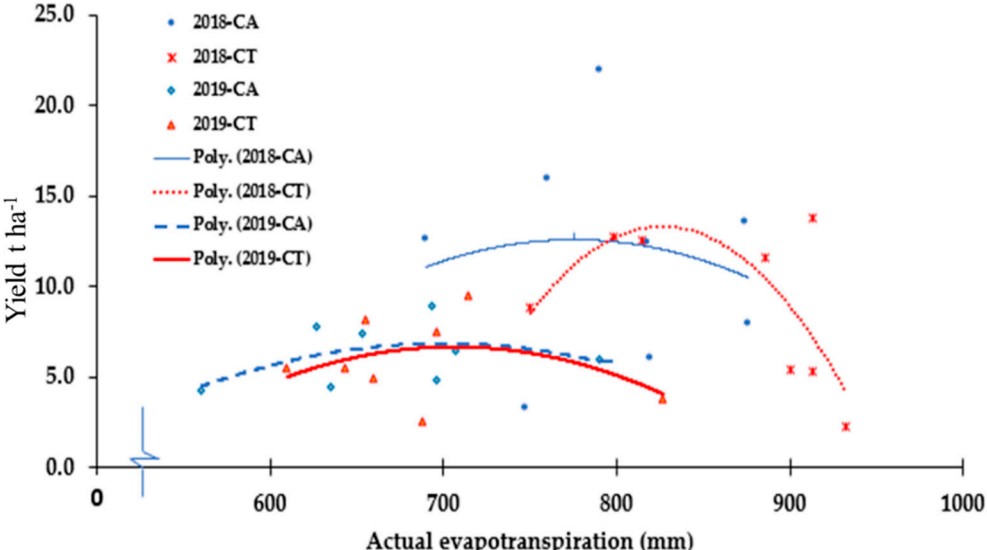

**Figure 5.** The relationship between evapotranspiration and pepper yield based on the data from replicated subplots of conservation agriculture (CA) and conventional tillage (CT) treatments conducted in 2018 and 2019 experimental years.

### 3.4. Nitrogen (NO$_3$-N) Dynamics

NO$_3$-N concentrations in leachate (percolated water) during the whole pepper growth period under CA and CT practices are shown in Figure 6. The concentration of NO$_3$-N in the leachate was greater in the CT management compared to in CA. At later crop growth stages, with an increased canopy cover, the difference in the concentration of NO$_3$-N between the treatments was mostly minimum (Figure 6). The mean concentration of NO$_3$-N in the leachate was 2.8 and 1.8 mg L$^{-1}$ in the CA treatment and 3.2 and 2.6 mg L$^{-1}$ in the CT, respectively for the 2018 and 2019 pepper growing seasons (Table 3). The mean NO$_3$-N loss in the leachate was significantly ($p < 0.05$) reduced under CA (29%) when compared with CT treatment (Table 3). Correspondingly, the load of NO$_3$-N in the leachate was 20.1 and 15.1 g ha$^{-1}$ in the CA treatment and 21.6 and 16.6 g ha$^{-1}$ in the CT for the cropping seasons of 2018 and 2019, respectively. When the amount of leachate decreased, the associated NO$_3$-N concentration increased at early crop stages over the drier months, i.e., from the start of cultivation to the harvest of pepper (Figure 6). Nitrate concentrations in the leachate for both treatments were the highest at the beginning of the growing period and decreased at the end of the growing period as rainfall amount increases (Figure 6).

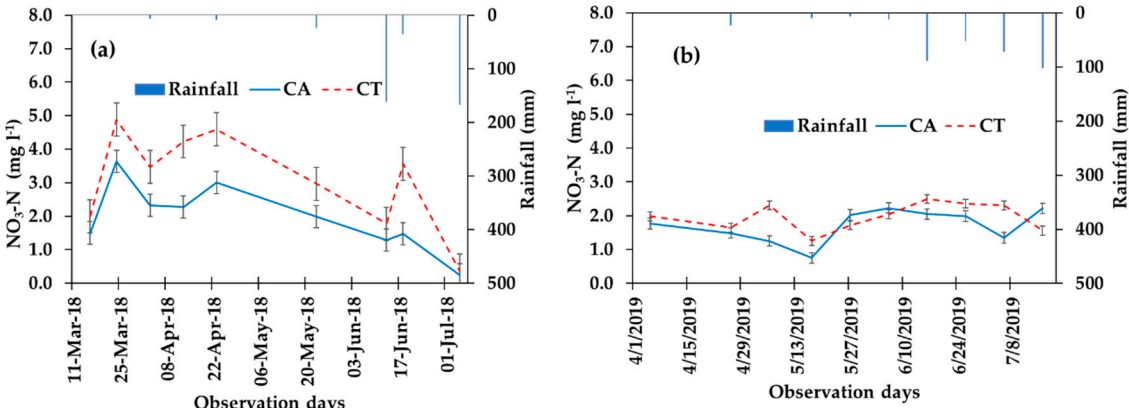

**Figure 6.** $NO_3$-N concentration (mg $L^{-1}$) in the leachate; data collected every 10 days during pepper production period under conservation agriculture (CA) and conventional tillage (CT) treatments in 2018 (**a**) and 2019 (**b**) experimental years.

**Table 3.** The mean concentration of nitrogen ($NO_3$-N) in the leachate (mg $L^{-1}$) and the corresponding load (g $ha^{-1}$), as well as the concentration of $NO_3$-N removed by surface runoff (mg $L^{-1}$) and the corresponding load (g $ha^{-1}$) for the two years in conservation agriculture (CA) and conventional tillage (CT) treatments.

| Variables | 2018 | | 2019 | |
|---|---|---|---|---|
| | **CA** | **CT** | **CA** | **CT** |
| $NO_3$-N (leachate), mg $L^{-1}$ | 2.8 ± 0.9 [b] | 3.2 ± 1.3 [a] | 1.8 ± 0.7 [b] | 2.6 ± 1.2 [a] |
| $NO_3$-N (leachate), g $ha^{-1}$ | 20.1 ± 7.8 [b] | 21.6 ± 9.1 [a] | 15.1 ± 12.8 [b] | 16.6 ± 16.2 [a] |
| $NO_3$-N (runoff), mg $L^{-1}$ | 0.3 ± 0.1 [b] | 0.6 ± 0.15 [a] | 0.4 ± 0.1 [b] | 0.8 ± 0.3 [a] |
| $NO_3$-N (runoff), g $ha^{-1}$ | 148.8 ± 66.2 [b] | 384.0 ± 75 [a] | 333.7 ± 122 [b] | 866 ± 359 [a] |

Numbers followed by the same letters under the same row heads in the same year are statistically nonsignificant at the $\alpha = 0.05$ significant level.

On the other hand, the concentration of $NO_3$-N in the runoff was 0.3 and 0.4 mg $L^{-1}$ in the CA and 0.6 and 0.8 mg $L^{-1}$ in the CT, respectively for the 2018 and 2019 cropping seasons (Table 3). Consequently, the load (g $ha^{-1}$) of $NO_3$-N in the surface runoff was found 39% lower in CA when compared with CT (Table 3). The result indicates that $NO_3$-N concentration was significantly ($p < 0.5$) lower in the runoff when compared with its concentration in the leachate.

*3.5. Phosphorus ($PO_4$-P) Dynamics*

Available phosphorus below the 40 cm soil layer showed a decreasing trend from the dry to wet months of the 2018 and 2019 cropping seasons. The concentration of $PO_4$-P decreased with an increase in the leachate in 2018 and 2019 (Figures 4 and 7). Figure 7 shows the changes in $PO_4$-P concentration over time in 2018 and 2019. The mean concentration of $PO_4$-P was 1.2 and 0.80 mg $L^{-1}$ in the CA and 0.8 and 0.6 mg $L^{-1}$ in the CT, respectively for 2018 and 2019 cropping seasons (Table 4). The mean $PO_4$-P concentration in the leachate in 2018 and 2019 was, respectively, 50% and 33% higher in CA as compared with CT, which is also significant ($p < 0.05$) (Table 4). Correspondingly, the load (g $ha^{-1}$) of $PO_4$-P in the leachate was 8.4 and 15.1 g $ha^{-1}$ in the CA treatment and 5.6 and 16.6 g $ha^{-1}$ in the CT, respectively for the 2018 and 2019 cropping seasons (Table 4). The $PO_4$-P concentration was higher at early crop stages while the quantity of leachate decreased in the dry months (March to May).

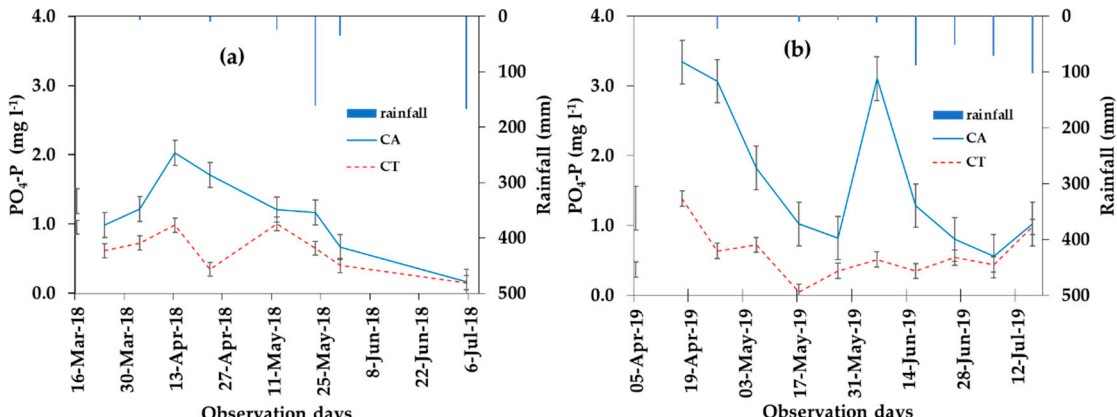

**Figure 7.** The concentration of $PO_4$-P (mg $L^{-1}$) in the leachate under CA and CT for 2018 (**a**) and 2019 (**b**).

**Table 4.** The concentration of phosphorus ($PO_4$-P) in the leachate (mg $L^{-1}$) and its load (g $ha^{-1}$), as well as the concentration of phosphorus ($PO_4$-P) removed by surface runoff in (mg $L^{-1}$) and the associated load (g $ha^{-1}$) for the two treatments and under experimental years.

| Variables | 2018 | | 2019 | |
|---|---|---|---|---|
| | **CA** | **CT** | **CA** | **CT** |
| $PO_4$-P (leachate), mg $L^{-1}$ | 1.2 ± 0.7 [a] | 0.80 ± 0.4 [b] | 0.80 ± 0.5 [a] | 0.6 ± 0.3 [b] |
| $PO_4$-P (leachate), g $ha^{-1}$ | 8.4 ± 4.0 [a] | 5.6 ± 2.6 [b] | 15.1 ± 4.2 [a] | 16.6 ± 2.6 [b] |
| $PO_4$-P (runoff), mg $L^{-1}$ | 0.55 ± 0.1 [a] | 0.64 ± 0.15 [b] | 0.6 ± 0.2 [a] | 0.7 ± 0.3 [b] |
| $PO_4$-P (runoff), g $ha^{-1}$ | 243 ± 66.2 [a] | 389 ± 75 [b] | 500.8 ± 215 [a] | 702.6 ± 312 [b] |

Numbers followed by the same letters under the same row heads in the same year are statistically nonsignificant at the $\alpha$ = 0.05 significant level.

On the other hand, the mean concentration of $PO_4$-P in the runoff was 0.55 and 0.6 mg $L^{-1}$ in CA and 0.64 and 0.7 mg $L^{-1}$ in CT for the 2018 and 2019 cropping seasons, respectively (Table 4). The difference between CA and CT in the mean concentration of $PO_4$-P in the runoff was statistically significant ($p < 0.05$) (Table 4). Similarly, the corresponding load (g $ha^{-1}$) of $PO_4$-P in the runoff was 243 and 501 g $ha^{-1}$ in the CA, whereas it was 389 and 702 g $ha^{-1}$ in the CT for 2018 and 2019 cropping seasons, respectively (Table 4), which is significant ($p < 0.05$).

## 4. Discussion

### 4.1. Effects of CA on Agricultural Water Management

Conservation agriculture (CA) showed reductions in irrigation water use and runoff while it has increased soil water/percolation in the root zone compared with conventional tillage (CT) (Table 2). This was potentially due to the protection provided by the mulch cover and due to the minimum disturbance of soil by no-tillage practices. The use of mulch reduces evaporation of water from the soil [32], reduces the runoff by absorbing the energy of raindrops, and increases the percolation of water by delaying the runoff [33]. No-tillage encourages less disturbance of soil pore networks and increases porosity, which then increases percolated water within the soil and reduces runoff [34]. This water flow within the root media again encourages an improvement in soil water use that reduces water stress of shallow rooted vegetable crops [20,35]. Irrigation water reduction of about 15% was reported under CA practice compared with CT in the dry phase of garlic production in similar growing conditions [27]. In agreement with this study, Babalola et al. [36] reported that vetiver grass mulch (2 tones $ha^{-1}$) decreased the runoff by 62% compared to the control (CT) while another study indicated the reduction in runoff by the use of crop residue mulches [37].

As discussed earlier, our objective with the CA study is not only to investigate the pathways of surface water (runoff) but also to observe the water movement within the soil profile under vegetable fields. In this study, we observed increased percolated water under CA compared with CT because of the use of mulch cover and no-tillage practices (Table 2). A continuous application of grass mulch cover prevented the formation of soil crust, which contributes to the reduction in surface runoff and the increases effectiveness of the macroporosity of the soil that enhances percolation water. Edwards et al. [38] observed that large numbers of continuous macropores formed by burrowing earthworms were observed in the no-till watershed compared with the tilled one, and the authors speculated that no-till contributed to high infiltration rates. Moreover, the increased grass mulch cover of no-till soil may produce a cooler and wetter environment near the soil surface, which is more favorable for micro flora and faunal activity [39,40]. Less soil compaction as a result of no-tillage combined with grass mulch directly encourages microbial activities and can improve the vertical water movement within the soil structure [24]. In addition, the primary concern with CA practices is not only to investigate the pathways of surface and subsurface water over or within the soil but also to observe the quality of the dynamic movement of water in the soils of irrigated vegetable fields. Understanding how CA and CT practices affect the movement of water, however, allows us to concentrate on the factors most likely to influence nutrient movement under supplementary irrigated farms.

### 4.2. Effects of CA on the Nitrogen Movement

Conservation agriculture practices reduced the concentration of $NO_3$-N in the leachate under a vegetable production system (Table 3), possibly due to grass mulch and no-tillage practices, both of which allow for minimum nutrient losses. More water applied in CT during dry irrigation months has probably increased the removal of $NO_3$-N by leaching due to fertilizer turnover by tillage (Table 2). At later crop growth stages, with an increased canopy cover, the difference in the concentration of $NO_3$-N between the treatments was minimum (Figure 6). The $NO_3$-N flux in the root environment was greater for some weeks after transplanting, while it decreased subsequently as the vegetative cover of pepper increased. In line with this study, a study in China showed that the $NO_3$-N concentrations in percolated water was in a regular decreasing pattern from drier to wetter phases of irrigated straw-mulched rice production [41]. Consistent result of $NO_3$-N load in the leachate was also reported by Govaerts et al. [42] in the CA experiment conducted in Mexico. Our study results are in line with the study in Croatia [21]. In both years, the concentration of $NO_3$-N in the runoff was lower in CA than in CT due to various possible reasons. In the context of CA, the method of fertilizer application and the minimum soil disturbance during crop cultivation were important since urea (46-0-0; N-P-K) fertilizer was locally applied to vegetables near the seedlings during the 1st irrigation phase. In this regard, more nutrient movement would be expected into the soil, not by the runoff. This is in agreement with the result of Yadav [43], which showed that 20% of the $NO_3$-N that joins the groundwater came from the root zone for most of the crops. Another study indicated that grass mulch incorporated greater soil organic matter and $NO_3$-N over surface soil layers, and this protected it from the runoff in the case of CA [44].

### 4.3. Effects of CA on Phosphorus Movement

In CA treatment, $PO_4$-P concentration (mg $L^{-1}$) and load (g $ha^{-1}$) were higher in the leachate and decreased in the runoff compared with the CT treatment (Table 4). The higher $PO_4$-P concentrations in the leachate may be attributed to its subsequent accumulation in the lower layers due to the higher water movement in the soil layer under CA practices. No-tillage combined with mulch, in general, is characterized by a higher phosphorus content in surface soil profile, which contributed to phosphorus dynamics within the soil layers compared to CT. Similar results have been also reported by Ben-Gal and Dudley [45]. A higher total phosphorus content was also reported in the soil surface layers under no-tillage compared to CT [46]. The higher phosphorus in no-tillage is mainly due to minimum soil disturbance, allowing for the accumulation of phosphorus fertilizer applied, as well as

the phosphorus of mulch or crop residues added though time [47,48]. No-tillage combined with grass mulch practices are also suitable for the transformation of inorganic phosphorus (P) added through the fertilizer into organic forms, thereby increasing biological P reactions in the soil surface layer [49]. $PO_4$-P concentration showed signs of decrease from the initial crop stage to harvest, in which case the concentration at each observation dates is higher for CA compared with that for CT (Figure 7). $PO_4$-P concentration decreased from the dry irrigation phase to the wet rainy period where rainfall was in excess. The reason for this may be the increase in nutrient uptake by the crop and the cumulative removal of $PO_4$-P by the runoff and leachate.

### 4.4. Effects of CA on Pepper Yield

Conservation agriculture, apart from numerous other advantages, improved yield and the early maturity of pepper compared with conventional tillage treatment (Figure 5), which is consistent with the results of Ravinderkumar et al. [50], showing that the application of organic mulches resulted in the flowering of tomatoes in fewer days after transplanting compared with the control management. In both years of this study, most of the initial and development stages of pepper were sufficiently supported by irrigation, and the fruit filing stages were supported by rainfall in both treatments. However, the yield achieved in CA treatment for 2018 and 2019 was higher compared with CT. The yield variation between the treatments was caused by the conducive soil moisture availability under CA management due to the use of grass mulch and minimum soil disturbance, particularly at the initial and development stages during the dry phase.

In 2019, the yield of pepper was lower than the yield in 2018, which may be attributed to the period of transplanting pepper relative to the rainfall onset. In 2019, transplanting was done one week later than 2018; it received more rainfall and was exposed to overwatering during fruit filing stage. As indicated before, the contribution of irrigation was higher in 2018 (46% for CA and 56% for CT) compared with 2019 (35% for CA and 37% for CT) for both treatments. In agreement with this study, Jaimez et al. [51] revealed that a water deficit and overwatering during the period of flowering and the fruit development stages reduced pepper fruit production. The authors, in addition, concluded that the transplanting of pepper about 2 months before a rainy season can improve the yield since the rain season coincides with flowering and fruit development stages [51]. Conversely, these stages are also critical, and water availability in the root zone in the dry phase is essential in order to avoid a significant decrease in fruit production, which was maintained by CA practices in this study. It has been observed that under CA practice, 20–40 mm additional water was stored in the root zone, especially in the lower root–soil layers, which is beneficial at the grain filling stage of the crop [27]. Wale et al. [52] also noted that the optimum crop water requirement of green pepper lies between 300 and 700 mm depending on the climatic conditions.

## 5. Conclusions

In this study, we observed that CA practices increased yields, reduced irrigation water, reduced runoff and the associated $NO_3$-N in the leachate and runoff compared to CT. The average of the two years indicated a 20% increase in yields, 21% decrease in irrigation water, 40% lower runoff, 38% higher percolation, 29% lower $NO_3$-N in the leachate, and 100% lower $NO_3$-N in the runoff under CA compared with CT. In contrast, $PO_4$-P concentration in CA was higher in the leachate but lower in the runoff. While N dynamics in the root zone indicated a decreasing rate with time, the phosphorus dynamics were not consistent across CA or CT. The observed responses of runoff, leachate, yield and nutrient dynamics under CA in this research provided a better understanding on the response of pepper to CA and the associated benefits in terms of increased yield as well as lower irrigation water and nutrient losses. We conclude that in supplementary irrigation vegetable production systems, CA can provide opportunities to optimize water use by decreasing irrigation water requirements and optimize nutrient use by decreasing nutrient losses through runoff and leaching. These best

management practices will not only improve yield and efficiency of inputs (water and nutrient) but also decrease pollution and protect our environment.

**Author Contributions:** S.A.B. has contributed to the conceptualization, methodology, investigation, data collection, and acquisition, data analysis, writing the original draft manuscript in scientific content. T.T.A. contributed to revising the manuscript for the intellectual content. P.S. contributed to the methodology and revising of the manuscript. A.W.W. and P.V.V.P. contributed to the revising and editing of the manuscript. T.S.S. contributed to the rewriting of the manuscript to the scientific content. M.R.R. contributed to the conceptualization and design of the experiment. S.A.T. contributed to the conceptualization, data collection, acquisition framework, supervision, project administration, funding acquisition and rewriting the manuscript to the scientific content. All authors have read and approved the final version of the manuscript.

**Funding:** This research was funded by the American people through support by the United States Agency for International Development (USAID) Feed the Future Innovation Lab for Collaborative Research on Sustainable Intensification (Cooperative Agreement No. AID-OAA-L-14-00006, Kansas State University) through Texas A&M University's Sustainably Intensified Production Systems and Nutritional Outcomes, and University of Illinois Urbana-Champaign's Appropriate Scale Mechanization Consortium projects. The contents are the sole responsibility of the authors and do not necessarily reflect the views of funding agencies or represented organizations.

**Acknowledgments:** This research is made possible by the generous support of the American people through support by the United States Agency for International Development (USAID) Feed the Future Innovation Lab for Collaborative Research on Sustainable Intensification. Additional support was received of the CGIAR Research Program on Water, Land and Ecosystems (WLE) supported by donors to CGIAR. The contents are the sole responsibility of the authors and do not necessarily reflect the views of USAID. The authors would also like to acknowledge the Blue Nile Water Institute and Bahir Dar Institute of Technology for sponsoring the project. Contribution number is 21-008-J from the Kansas Agricultural Experiment Station.

**Conflicts of Interest:** The authors declare no conflict of interest.

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
