# Peer review of "The Response of Water and Nutrient Dynamics and of Crop Yield to Conservation Agriculture in the Ethiopian Highlands"

_sustainability, doi:10.3390/su12155989_

Round 1
Reviewer 1 Report
The paper “The response of Water and Nutrient Dynamics and Crop Yield to Conservation Agriculture in the Ethiopian Highlands” is resubmitted version of previously rejected paper in Sustainability Journal.
Overall, the paper is significantly revised where the majority of the text is rewritten. The authors have addressed and revised their manuscript according to my comments or not and it seems that paper have much better quality. Especially discussion section. However, the general complaints look the same as previous:
1) novelty is poorly explained. In other words,… I am not convinced. Authors stated “However, experimental field measurements on water and nutrients dynamics and associated pepper yield are missing for a comprehensive impact analysis of CA.” I believe that the pepper is vegetable on cropland, and many studies about water and nutrient dynamics on cropland (together with crop yields) exist. If the pepper yield is reason for novelty, then I apologies immediately.
2) “The results are quite expectable, basically, where the vegetation cover is greater and soil disturbance lower, the runoff and nutrient losses are smaller. Therefore, the contribution of the manuscript to the knowledge of soil behavior under different management systems is very limited.” This grade is written in first version and stay also like grade in second version.
If we ignore this two fact, the paper does not have many other faults in this new version. It is written very well.
Sincerely
Minor things:
Line 116: state type of tiilage – invertive (plow), non-invertive – cultivator/chisel/ripper
Line 119: please check the proper way of citing again
Line 137: State is it manure or compost. I believe that is compost according to organic matter concentration
Line 182. Divide “rise(mm)”
Line 226: here and elsewhere. Please use terms significantly or not significant when you describe your results.
Line 235: please divide “4).The“.
Figure 3 differ in font letter between “a” and “b” legend and in font of y-axis titles.
Line 272-275. Please be consistent in writing. Somewhere you put “/” (mg/L) somewhere “-1” (g ha-1). Please unify here and elsewhere. Check all manuscript.
Line 275-276. Please put the units after the numbers.
Line 341: Please remove comma at “Edwards. et.al. [38]”
Line 374: put L instead l.
Line 408: Here and elsewhere. Somewhere authors put “reference [52]”, somewhere last name of the reference and number in brackets. Please unify. Check the texts.
Author Response
Dear reviewer:
I hope i addressed all the issues and improved the manuscript.
thank you.

Reviewer 2 Report
Authors have successfully replied to most of previous comments. However, I still found some issues requiring authors' attention. Please check the attached pdf.

Author Response
Dear reviewer:
i hope I addressed all the comments and the revised manuscript is well improved.
thank you.

This manuscript is a resubmission of an earlier submission. The following is a list of the peer review reports and author responses from that submission.
Round 1
Reviewer 1 Report
The manuscript explores the influence of soil management systems on some properties of soils, runoff and N/P leaching, what is a relatively well-known subject. The authors justify their work arguing that no similar study was carried out in the Ethiopian highland, (lines 75-78), but will the soil behave in a different form under similar environments?
The results are quite expectable, basically, where the vegetation cover is greater and soil disturbance lower, the runoff and nutrient losses are smaller. Therefore, the contribution of the manuscript to the knowledge of soil behavior under different management systems is very limited. In addition to the mentioned problem there are other defects in the manuscript. Introduction section missing previous findings of runoff on conventional, and CA plots. Also, this the same on previous findings about N and P loss in runoff. I do not see proper description of identification of the gap in knowledge, moreover I do not see novelty except the statement that such studies missing on regional level. At the end of introduction hypothesis missing, while novelty I found very weak. Methods section missing information about the management. Readers from abroad of Africa do not familiar with “Oxen” as the system. State type of tool used and depth of tillage. The number of replications on one site is unknown. Please state the number of repetitions on each of 10 sites.
Results section should be improved. Many inconsistences appear in way of data presentation. Data should be presented clear without additional explanations and discussions. Here I draw attention of horrible graphic solutions. Graphic should be unified, better quality, without typograph errors and different fonts. Authors should also avoid doubling the data presentations in tables and graphs.
Discussion section is weak. In many parts discussion missing. Authors to many repeat the results and shows the results of different studies instead they discuss their own results. Moreover, they need to avoid speculations with properties that was not measured in this work.
Conclusion should show the essence and take-home message of the manuscript. Highlight the most important data, derive the conclusion from it. Do not write the conclusion like the summary of the paper. Write conclusions in order to answer the hypothesis and aims of the paper (which you need to add in revised version). One go-home message and close this part. Please avoid conclusions derived from not-measured properties. Please rewrite
Minor things are below
Sincerely
Line 50-53: Please add supportive citation. On present way is written like speculation.
Line 62: Please use full word before first time you use abbreviation.
Line 67: This is very disputable fact. Many studies confirm opposite. Please state that this could appear depending on initial soil state, management and duration of transition to no-till management. Moreover, numerous studies report similar or lower crop yields on no-tillage, so please state that this finding occur on ?? environment and ?? soil.
Also, very important, avoid to cite dissertations, technical reports and papers published in conference books.
Line 68: Please pay attention during citing articles. Both papers are nor related to no-tillage, only the first have something with much, but with PE, not grass mulch. Rewrite the sentence or find other studies. Cheech all citations.
Line 85-86: Please state the source of the weather data.
Line 104: Please write full word before abbreviation, please. Authors should describe what CT in the agroecosystems of Ethiopia highland means?
Line 129-130: Only on CA plots? Please state it.
Table 1: Please state type of tool and depth of tillage
Line 136-137: This have to be highlight in text, not as table description.
Line 153-154: From the figure 1 is not possible to distinguish number of plots on each treatment. Are there 4 plots but only one collector?
Line 167: Please be consistent in type of writing. Put NO3-N.
Line 183: Write full name for WS method.
Line 199: Figure 2 need adjustment. Is not visible to check what is written. Remove the outline borders, unify the size of both graph. Mark each graph by letter and describe it in the figure title.
Table 2: As I understand statistic, "a" always go near highest number during comparisons. Please check all manuscript.
Line 216: This cannot be seen from the figures. There are no letters used for statistics. Moreover, the duplication of the presented results is not advisable. This data was already presented in Table 2. Please delete figure 3.
Line 226: Figures are poor. Need to be better quality. You have different y axis title. Correct this. Unify the size of graphs, remove the outline border.
Line 229: You mean on figure 5?
Line 238: Type of figure should be like in previous ones. Separate for 2018 and 2019. Same for figures 7 and 8.
Line 246: Results of pepper yields should be presented in t/ha or kg/m2.
Line 302: Here authors should discuss the results in similar order like they presented them. Please before you start with hydrological response, discuss the soil water conservation differences between plots.
Line 303-322: Please explain to readers what happen with mulch and how the mulch affects the soil structural characteristics (infiltration, macrospores, etc.). State the findings of other on similar textured soils.
Line 324-325: I believe that tillage management is the crucial factor for this....
Line 327-328: Please relate this with crop N uptake in different stages of development and give examples from other studies.
Line 336: Please, as I stated before, pay attention about citations. Romic et al was study conducted in Croatia, not India.
Line 336-337: Find studies focused on runoff and vegetation cover. Relate this with runoff concentrations and N losses.
Line 345-354: Authors should less repeat the results and what other persons find and more try to discuss their own data.
Line 365-367: This is very speculative fact. Authors did not study soil properties as supportive variables.
Line 376-378: Only this? Please relate the yields with possible soil state (write it hypothetically using "probably", "very likely" etc.) and nutrient losses on each treatment. Use also reports of others in different countries but try to find similar textured soils.
Reviewer 2 Report
Review of manuscript sustainability-834567, entitled “The response of water, nutrient, and crop growth dynamics to conservation agriculture in the Ethiopian highlands” by S.A. Belay, T.T. Assefa, P. Schmitter, A.W. Worqlul, T.S. Steenhuis, M.R. Reyes, and S.A. Tilahun.
The paper compares the effect of conservation agriculture and conventional tillage on the soil water balance and nutrient (nitrate and phosphorus) leaching and runoff of 10 experimental fields located in Northern Ethiopia. The content of the paper is certainly interesting, but it has raised me many doubts when reading it. Please see my comments below as well as the attached pdf document where I’ve enclosed some additional notes.
- The introduction section needs to be revised. There are some sentences that are based on wrong concepts. The notion that the source of phosphorus and nitrogen is rainfall and irrigated agriculture is strange (L49-50). This needs to be better explained. Authors also state that no-till practices lead to water savings (L67-68), but they don’t refer explain what they mean by that. Is it increased water retention in the soil profile? Is it improved irrigation scheduling?
- The description of the experimental treatments need to be more clearly presented. Authors really need to avoid sentences like “Rainfall, water use, leachate, and nutrient transport data were collected in 2017, 2018 and 2019” which is totally vague when no extra information is given in this section. A better organization of the paper may be needed.
- Authors used a wetting front detector to assess percolation (L108). I’m not familiar with this kind of equipment but apparently it works like a lysimeter. Authors need to explain its functioning in detail.
- Details of the irrigation experiment need to be given. How much water was applied and the frequency?
- Authors need to revise their concepts related to the soil water balance and determination of evapotranspiration. There as some serious errors here as pointed out, for example, in L113-114, L144-150, and L159-160.
- The experiment was run from 2017 to 2018, but the weather data is oddly from 1995-2016.
- Nutrient losses should be presented in terms of mass and not concentrations which are dependent on the amount of water in runoff and leaching. If given in terms in mass we can better assess whether conservation agriculture improved nutrient efficiency or not.
